# SkipDecode: Autoregressive Skip Decoding with Batching and Caching for Efficient LLM Inference

## Abstract

Autoregressive large language models (LLMs) have made remarkable progress in various natural language generation tasks. However, they incur high computation cost and latency resulting from the autoregressive token-by-token generation. To address this issue, several approaches have been proposed to reduce computational cost using early-exit strategies. These strategies enable faster text generation using reduced computation without applying the full computation graph to each token. While existing token-level early exit methods show promising results for online inference, *they cannot be readily applied for batch inferencing and Key-Value caching*. This is because they have to wait until the last token in a batch exits before they can stop computing. This severely limits the practical application of such techniques. In this paper, we propose a simple and effective token-level early exit method, `SkipDecode`, designed to work seamlessly with batch inferencing and KV caching in autoregressive text generation scenarios . It overcomes prior constraints by setting up a singular exit point for every token in a batch at each sequence position. It also guarantees a monotonic decrease in exit points, thereby eliminating the need to recompute KV Caches for preceding tokens. Rather than terminating computation prematurely as in prior works, our approach bypasses lower to middle layers, devoting most of the computational resources to upper layers, allowing later tokens to benefit from the compute expenditure by earlier tokens. Our experimental results show that `SkipDecode` can obtain 2x to 5x inference speedups with negligible regression across a variety of tasks. This is achieved using OPT models of 1.3 billion and 6.7 billion parameters, all the while being directly compatible with batching and KV caching optimization techniques.

## 1 Introduction

Autoregressive large language models (LLMs), such as the GPT Radford & Narasimhan (2018) and OPT Zhang et al. (2022) family, have demonstrated strong performance across a wide range of tasks Radford et al. (2019); Raffel et al. (2020); Brown et al. (2020). However, they also have high computational cost and latency requirements resulting from token-by-token generation. Token-level early exit Schuster et al. (2022); Sun et al. (2022) has emerged as a promising technique to alleviate these limitations by allowing tokens to cease computation as soon as their hidden states reach saturation Schuster et al. (2022).

Although current methodologies exhibit theoretical advantages, their practical implementation is somewhat restricted since they are not compatible with batch inferencing and KV caching techniques, which are widely used to speed up inference in practice. This is mainly due to the necessity of prolonging computation until the final token in a batch for each position is thoroughly processed. This effectively limits improvements to the exit position of the most computation-heavy token. Additionally, token-level exit strategies, which depend on dynamic indicators like learned classifiers for defining exit points, don't provide any assurance concerning computational expenses, such as the worst-case cost relative to the computation performed by the full-sized network. A further practical difficulty arises in the form of Key-Value (KV) caching of prior tokens, which requires updating if the current token exits later than the others.

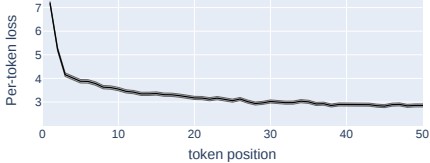
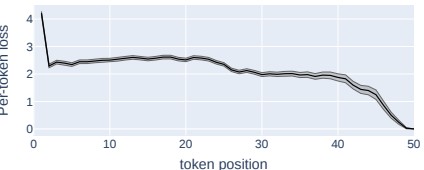

(a) **OPT-350m** on **OpenWebText**. Average loss per token position shows a strong monotonically-decreasing trend for general text.

(b) **OPT-350m (finetuned)** on **Reddit-TLDR**. Average loss per token position. Decreasing trend but with a different function.

Figure 1: Average loss per token position (black) during the forward pass of OPT-350m model on a general and a task-specific dataset. Grey represents the 95% confidence interval on the mean.

In this paper, we present a novel token-level early exit method focused on the autoregressive text generation step, called `SkipDecode`, which overcomes these limitations, while maintaining a controlled computational budget. Our approach establishes a unified exit point for all tokens within a batch at a specific sequence position. We further capitalize on the observation that words towards the end of a sequence are generally easier to predict due to more contextual information.

Previous work Schuster et al. (2022) also supports our hypothesis illustrating that early incorrect predictions bear a significant cost compared to later mistakes in the sequence. They also show that initial tokens in the sequence are harder to approximate using fewer layers compared to latter tokens. Figure 1 complements this data by showing the decreasing loss per token position: Predictions at the beginning of the sequence register higher entropy in contrast to the tokens that appear later. This insight inspired our generation policy with monotonically decreasing exit points as the sequence progresses, with the assumption that subsequent tokens demand less computational effort. The idea is to use increased computational effort upfront to minimize mistakes, and therefore prevent the cascading of errors. Our strategy also eliminates the necessity to recalculate Key-Value (KV) caches for preceding tokens, substantially reducing computational cost.

Tokens exiting at distinct layers are unable to benefit from all the information generated by previous tokens that exit at later positions, leading to wasted computation and loss of contextual information. To address this issue, the early exit mechanism in `SkipDecode` leverages the entire computation performed by all tokens, resulting in a substantial improvement in the speedup-task performance trade-off. Rather than abruptly ending computation, our approach bypasses lower layers and primarily allocates the computational budget to upper layers, enabling rightward tokens to benefit from the computational resources employed by leftward tokens effectively.

Our technique `SkipDecode` (overview in Figure 2 and Table 1) is able to avoid performance degradation up to the hidden state saturation point. We experiment with up to 5x speedup over 1.3 billion, and 6.7 billion OPT models on three generation datasets. We also solve practical problems like batching and KV caching while maintaining a controlled and predictable computational budget. Our method makes it easier to use LLMs with limited resources and helps to democratize AI.

| Method | Generation | Token Level | Batching | KV-Caching | Full Attention | Controlled Comp. Cost |
|---|---|---|---|---|---|---|
| CALM | ✓ | ✓ | ✗ | ✗ | ✗ | ✗ |
| SkipDecode | ✓ | ✓ | ✓ | ✓ | ✓ | ✓ |

Table 1: Comparison of CALM and `SkipDecode`. `SkipDecode` supports batching and KV caching for increasing inference efficiency with controlled computational budget.

## 2 ADDRESSING PRACTICAL CHALLENGES IN TOKEN-LEVEL EARLY EXIT STRATEGIES

### 2.1 OPTIMIZING BATCHED INFERENCE WITH UNIFIED EXIT POINTS

Batched inference is widely used to enhance computational efficiency by simultaneously processing multiple input samples. This approach capitalizes on the parallelism offered by hardware, such as GPUs and TPUs, to reduce latency and improve throughput.

Practical Blockers (Existing):

- **Batching:** Computational cost defined by the last exit token

- **KV Caching:** If next token exits later than previous one, we need to recompute KV values for previous tokens.

- **Computational Efficiency:** If next token exits earlier, it does not attend full computation of previous token.

- **Cost Uncertainty:** Worst case scenario (for a bad token exit, e.g., from last layer) equivalent to processing the whole network.

Solutions (Ours):

- **Batching:** Exit per position per batch (column-wise).

- **KV Caching:** Next column has to exit earlier than previous column. Leftwards tokens are more difficult to generate.

- **Computational Efficiency:** Spend most of computational budget on top layers. Implicitly attends the full computation of previous tokens.

- **Cost Uncertainty:** Static policy (no surprises), computation costs are predefined.

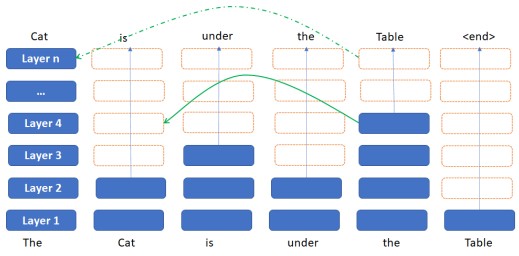

(a) Early Termination

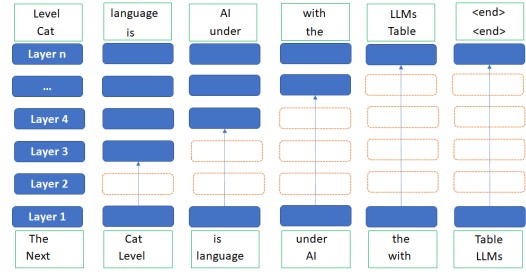

(b) Skipping

Figure 2: Overcoming practical blockers in token level early exit for language generation.

However, when applying existing token-level early exit strategies Schuster et al. (2022) to batched inference, challenges arise primarily due to the varying exit points of tokens within a batch. Given that tokens exit at diverse layers, it's necessary to persist computation until the final token of each batch member and each position is processed. This diminishes benefits undermining the potential advantages of parallel computation.

To tackle this, our method designates a fixed positionwise exit point for every token in a batch at a given sequence position. This strategy ensures that the processing of all batched tokens at a certain sequence position concludes at the same time. As a result, it assures that all theoretical benefits observed during evaluation are fully actualized during generation for non-trivial batching scenarios.

Let $B$ be the batch size, and $N$ the sequence length. We construct batches column-wise using tokens at a specific position across instances. Considering $t_{s,n}$ to be the token in sequence $s$ at position $n$, a given batch consists of tokens from $t_{(\cdot),n}$. Let $L(t_{b,n})$ be the layer at which token $t_{b,n}$ exits. We ensure that $\forall n \in [1, N], \forall b_1, b_2 \in [1, B], L(t_{b_1,n}) = L(t_{b_2,n})$. Further, the autoregressive decoding ensures that the columns are processed left to right such that the computation for tokens at position $n$ can utilize all the network computation from processing those at the previous position $n - 1$.

## 2.2 KV CACHING AND MONOTONICALLY DECREASING EXIT POINTS

Key-Value(KV) caching is a critical optimization technique for efficiently executing attention mechanisms in Transformer models. By storing the computed keys and values for previously processed tokens, the model can significantly reduce redundant computations when attending to the same context during subsequent steps. This optimization leads to faster inference times.

Yet, when utilizing token-level early exit strategies , the different exit points of tokens in a sequence present another challenge. Specifically, there's a requirement to recompute the Key-Value (KV) caches for preceding tokens if the current token exits at a higher layer. This necessary recalculation escalates computational workload and undermines the advantages of early exit methods, as the computation of each preceding token is bounded by the computation of later tokens.

Our proposed solution, which assigns a unified exit point to all tokens in a batch at a given position, effectively addresses this challenge. By ensuring that batched exit points are monotonically decreasing as the sequence progresses, we guarantee that previous tokens have performed at least as much computation as the current one, thereby trivially avoiding the need for any extra computation. The right plot in figure 2 shows how every layer can attend to leftward attention layers without any re-computation or change in the architecture.

The underlying rationale is that next-word prediction at the beginning of a sequence is more challenging due to limited context, and therefore earlier tokens will benefit from later exit points. Prior work Schuster et al. (2022) have already showed that noise or perturbations have a greater impact

on the overall task performance when the perturbation happens in the earlier tokens resulting in cascading of errors due to autoregressive generation. Moreover, as the context grows with the sequence, the later hidden states saturate faster (i.e. hidden states have limited variance across layers). Therefore, later tokens require less computation and enabling a more efficient use of computational resources Holtzman et al. (2019). We demonstrate this intuition in Figure 1, where earlier tokens in a sequence have higher losses and are more difficult to generate in contrast to the ones appearing later that are more predictive. An added benefit of our method is the reduction in cache size, which in turn allows room for potentially larger batch sizes.

## 2.3 CONTROLLING COMPUTATIONAL BUDGET WITH BATCHED EXIT POINTS

Traditional early exit techniques typically learn exit points for individual tokens Schuster et al. (2022). However, apart from the limitations mentioned in the previous subsections, controlling the computational budget can be challenging. Usually, a classifier is used to decide whether a token should exit at a specific layer, resulting in the worst-case computational budget scenario being close to the cost of using the full network (for instance, bad exit point close to the last layer).

We address this issue by pre-specifying maximum and minimum exit points (the maximum and minimum number of layer that each token should go through), which controls the computational cost via the number of active model parameters. Exit points across the sequence are assigned in such a way that no token exceeds the maximum nor falls below the minimum exit point keeping the total computational cost bounded. Additionally, as explained earlier, the assignment of exit points across the sequence is required to be monotonically decreasing. This implies that the first token will be assigned the maximum exit point, and the last token, according to the maximum length parameter, will be assigned the minimum exit point.

A predefined function progressively designates exit points to the tokens in the sequence. This function can adopt multiple forms and serves as an additional hyperparameter of the model, managing the computational expenditure. In the evaluation phase, we conduct tests using linear decay bounded by a a minimum and a maximum number of layers. Note that employing other functions (such as power-law) could lead to more significant accelerations and will be the subject of future studies.

Formally, consider a sequence and network with hyper-parameters: sequence_length, min_exit_layer, max_exit_layer, num_decoder_layers, and prompt_size. We define an array token_idx as:

$$\text{token\_idx}[i] = \begin{cases} \text{num\_decoder\_layers} & \text{if } i < \text{prompt\_size} \\ (1 - t_i) \times \text{max\_exit\_layer} + t_i \times \text{min\_exit\_layer} & \text{if } i \geq \text{prompt\_size} \end{cases}$$

where $t_i = \frac{i - \text{prompt\_size}}{\text{sequence\_length} - \text{prompt\_size}}$.

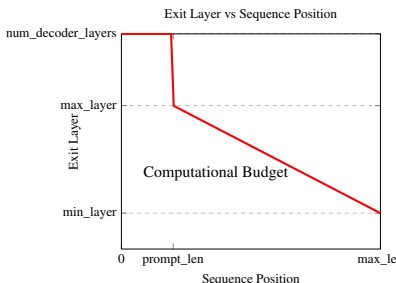

Figure 3: Linear decay of exit layer with respect to the sequence position.

In the above design, we can process all the tokens in the prompt with the full computational power of the network (i.e. using all the decoder layers). This can be done efficiently with batching since there is no generation involved in prompt processing. As soon as we start autoregressive token generation from prompt_len $+ 1$, we start decaying the number of active layers acting on any token bounded by the max and min layers pre-specified by the computational budget.

The overall cost is intrinsically linked to the sequence length. Typically, a maximum generation length is set, corresponding to the datasets or model's specifics. However, if generation optionally

exceeds this length, then the exit tokens are assigned to the model's minimum layer. During training, we employ the median prompt length found in the dataset's training data. This choice provides adaptability during testing, ensuring that samples with diverse prompt lengths can be effectively handled.

## 2.4 SKIPPING VS. EARLY TERMINATION

Early termination-based methods allow tokens to exit the model early once their hidden states have saturatedSchuster et al. (2022). However, token-level early termination can present problems in generative models when previous tokens exit earlier than the later tokens. In this scenario, the later tokens are unable to benefit from the extra computation performed by the former tokens via the attention mechanism, effectively under utilizing the available context.

To overcome this limitation, we propose performing skipping instead of early termination. We ensure that the computational budget for each token is allocated to higher layers of the model. Now regardless of the exit points, tokens will be able to attend to the top layers of all previous tokens effectively attending to all the available context.

The inherent layer-by-layer hidden states incremental updating of transformers results in states gravitating towards saturation as computation advances Schuster et al. (2022). To counteract a potential disparity in representation, we introduce "warm-up layers". These layers denote the preliminary computation done on the initial $x$ layers before transitioning to the uppermost $y$ layers. Through testing, we observed that this method effectively bridges the gap between initial token embeddings and the apex layer's hidden states. Empirical testing on our datasets consistently revealed that 1 warm-up layer was most effective across various configurations.

| Original / Base Number of Layers | Target Speedup ($\times$) | #Target Avg Layer | #Warm up Layer | #Min Layer | #Max Layer |
|---|---|---|---|---|---|
| 32 (6.7B) | 2 | 16 | 1 | 11 | 22 |
| | 3 | 11 | 1 | 8 | 14 |
| | 4 | 8 | 1 | 6 | 10 |
| | 5 | 6.5 | 1 | 5 | 8 |
| 24 (1.3B) | 2 | 12 | 1 | 8 | 16 |
| | 3 | 8 | 1 | 6 | 10 |
| | 4 | 6 | 1 | 5 | 7 |
| | 5 | 5 | 1 | 4 | 6 |

Table 2: `SkipDecode` configurations for different target speed-ups w.r.t Base OPT (1.3B and 6.7B) obtained using the E2E validation set corresponding to the least perplexity.

## 3 EVALUATION

We demonstrate our techniques with OPT Zhang et al. (2022) decoder-only language models of 1.3b and 6.7b parameters (24 and 32 layers respectively) on three text generation datasets: E2E Novikova et al. (2017), Reddit-TLDR Völske et al. (2017), and CNN-DM Hermann et al. (2015). Together, these datasets encompass structured data translation, user-generated content and news article summarizing, offering a comprehensive evaluation of our techniques. We implemented `SkipDecode` using the metaseq codebase[1].

### 3.1 EXPERIMENTAL DESIGN AND DATASETS

Given a base model assigned a $1\times$ speedup, our goal is to reduce the amount of computation performed by the network during inference using `SkipDecode`. We evaluate our method with configurations corresponding to $2\times, 3\times, 4\times$ and $5\times$ target speedups. **Note that speedups are reported relative to the base model that intrinsically supports batching and KV caching.** This speedup comparison is different from prior early-exit works that consider a weaker base model with a batch size of 1 and no KV caching.

Different configurations of the maximum and minimum layer per token, as outlined in Section 2.3, can achieve different speedup. We determine the optimal combination of maximum and minimum layers,

---

[1]https://github.com/facebookresearch/metaseq

along with the warm-up layers and learning rate, for each specified speedup through hyper-parameter tuning on the e2e dataset based on the perplexity metric on the validation set.

It should be noted that the actual speedup may slightly vary during generation as it's impossible to predict in advance the number of tokens that the model will generate. However, the computational budget is strictly bounded by the minimum layer assigned to the maximum sequence length for each dataset in the train set. If generation goes beyond it, the exit layer of tokens going the maximum sequence length will be the minimum. The configurations used are presented in Table 2.

For fine tuning, we used the median training prompt length from each dataset for all instances, ensuring that all layers are processed to mimic the desired generation behavior as illustrated in figure 3. For each configuration, the base model is a pretrained one that undergoes fine-tuning, incorporating specific additional hyperparameters—namely min-layer, max-layer, warm-up layer, median prompt length, and maximum sequence length. Notably, the $1\times$ model represents the originally fine-tuned version where both min-layer and max-layer correspond to the model's total number of layers.

It's worth noting that our approach is effective yet simple and easy to implement. Besides the token skipping policy, it does not necessitate any additional modifications to the transformer architecture, either during training or generation.

| Dataset | Size | Target Speedup | #Target Avg Layer | #True Avg Layer | True Speedup | Bleu | Rouge-L | Bert-F |
|---------|------|----------------|-------------------|-----------------|--------------|------|---------|--------|
| E2E | 1.3b | 1 | 24 | 24 | 1.0 | 65.8 | 67.6 | 70.3 |
| | | 2 | 12 | 14.7 | 1.6 | 66.3 | 67.9 | 67.8 |
| | | 3 | 8 | 9.4 | 2.6 | 66.3 | 68.1 | 67.3 |
| | | 4 | 6 | 6.8 | 3.5 | 65.6 | 66.8 | 66.5 |
| | | 5 | 5 | 5.8 | 4.1 | 64.2 | 66.3 | 65.2 |
| | 6.7b | 1 | 30 | 30 | 1.0 | 64.2 | 66.6 | 70.8 |
| | | 2 | 15 | 20.3 | 1.5 | 65.3 | 68.2 | 67.6 |
| | | 3 | 11 | 13 | 2.3 | 65.9 | 68.0 | 67.7 |
| | | 4 | 8 | 9.4 | 3.2 | 66.9 | 67.9 | 67.1 |
| | | 5 | 6.5 | 7.6 | 3.9 | 64.0 | 65.7 | 65.2 |
| Redit | 1.3b | 1 | 24 | 24 | 1.0 | 9.0 | 27.3 | 31.9 |
| | | 2 | 12 | 15.6 | 1.5 | 8.9 | 27.5 | 32.1 |
| | | 3 | 8 | 9.9 | 2.4 | 7.0 | 25.1 | 22.9 |
| | | 4 | 6 | 6.4 | 3.8 | 3.9 | 21.3 | 11.5 |
| | | 5 | 5 | 5.0 | 4.8 | 3.0 | 19.6 | 7.1 |
| | 6.7b | 1 | 30 | 30 | 1.0 | 9.6 | 28.3 | 33.7 |
| | | 2 | 15 | 19.8 | 1.5 | 9.3 | 27.7 | 32.3 |
| | | 3 | 11 | 13.7 | 2.2 | 8.0 | 26.0 | 25.3 |
| | | 4 | 8 | 9.4 | 3.2 | 5.2 | 21.3 | 9.3 |
| | | 5 | 6.5 | 6.5 | 4.6 | 4.0 | 19.3 | 7.4 |
| CNN-DM | 1.3b | 1 | 24 | 24 | 1.0 | 15.8 | 29.5 | 35.9 |
| | | 2 | 12 | 15.6 | 1.5 | 15.0 | 28.9 | 34.8 |
| | | 3 | 8 | 8.9 | 2.7 | 7.8 | 23.3 | 20.2 |
| | | 4 | 6 | 6.2 | 3.9 | 3.2 | 18.6 | 2.3 |
| | | 5 | 5 | 5.3 | 4.5 | 4.0 | 18.1 | 2.5 |
| | 6.7b | 1 | 30 | 30 | 1.0 | 16.3 | 30.2 | 37.1 |
| | | 2 | 15 | 21.3 | 1.4 | 15.2 | 29.6 | 35.9 |
| | | 3 | 11 | 11.8 | 2.5 | 4.8 | 21.8 | 17.9 |
| | | 5 | 6.5 | 6.9 | 4.3 | 4.6 | 18.5 | 2.7 |

Table 3: `SkipDecode` performance on different datasets for varying speedups and base model sizes.

We performed experiments on three datasets. Examples of generation on each dataset are shown in Appendix, Table 5. For generation, in all cases we employed a beam of 1, top-sampling of 0.7, and a temperature of 0.3. For training we started from a pretrained checkpoint and fine tuned with 200 warm-up updates and swept over learning rates in the range of 2e-4 to 8e-6.

**E2E Novikova et al. (2017).** The task is to convert structured information from key-value pairs into fluent text. It is relatively small, comprising of 42061 training samples, 4672 evaluation samples, and

4693 test samples. The median prompt contains 38 tokens. We set a maximum sequence length of 160 and a maximum prompt length of 60, both for training and generation. The effective batch size is 256. We use a breakline to separate the prompt and completion with 650 warm-up steps and 8 epochs.

**Reddit-TLDR Völske et al. (2017).** A summarization dataset that includes a training size of 117,000 samples, an evaluation size of 6450, and a test size of 6550 samples. The median training prompt is 348 tokens long. We utilize 200 warm-up steps, an effective training batch of 32, and 3 epochs. The maximum prompt length is set at 512 tokens, and the maximum sequence length at 1024 tokens. The separator between context and completion is "\nTl;dr\n".

**CNN Daily Mail Hermann et al. (2015).** Requires writing a summary given an input article. A large dataset with a training size of 287,113 samples, an evaluation size of 13,368, and a test size of 11,490 samples. The median train prompt length is 788 tokens. We set the maximum sequence length at 2048 and the maximum prompt length at 800. The warm-up updates are set to 650, the effective batch size is 32, and we train for 2 epochs. The separator between context and completion is "\nTl;dr\n".

## 3.2 KEY RESULTS

Results are presented in Table 3. `SkipDecode` demonstrates significant improvement in computational efficiency for each dataset and model size in the autoregressive step. To measure computational cost we use the average layers used per token. Then, speedup is calculated as the maximum number of layers divided by the layers used. Configurations of the different hyperparameters to achieve a given target speedup (i.e., max-layer and min-layer) are calculated according to dataset specifics of the training set (i.e. max sequence length, and median prompt length). However, as the model may be generating different length sequences on the test set the target and actual speedup may differ.

When we establish a maximum potential length, we're essentially predicting a worst-case scenario. If the sequences turn out to be shorter than this hypothetical maximum, the actual speedup is slightly worst. If the generation goes beyond, each extra token will be assigned the minimum exit point.

As depicted in Figure 4 in Appendix, there is no noticeable performance degradation from $1\times$ (base model) to $2\times$ speedup, after which there is steady decline in performance with increasing speedups. We hypothesize that this is due to the tokens reaching the hidden state saturation point, beyond which further computation reduction leads to performance degradation. This pattern is consistent across datasets. We notice a delayed degradation in E2E, while CNN-DM starts to degrade more quickly given its relative difficulty.

**E2E** `SkipDecode` As the target speedup increases from $1\times$ to $5\times$, the average number of decoder layers active in the generation process reduces, signifying a decrease in the computation load. Interestingly, all the task measures corresponding to the Bleu, Rouge-L, and Bert-F scores remain relatively steady with a minor decline as the target speedup increases. This indicates that our method can accomplish significant speedups with minimal degradation for certain task settings.

**Reddit** Similar to the others, the average generation layer decreases as the target speedup increases. However, the performance metrics such as Bleu, Rouge-L, and Bert-F scores display more significant reductions compared to the E2E dataset, given the relative difficulty of this task. Wile our method still achieves significant speedups, the trade-off in terms of task performance is more noticeable.

**CNN-DM** The results follow a similar trend to the previous: as the target speedup increases, the average generation layer decreases, indicating reduced computational requirements. However, the performance metrics such as Bleu, Rouge-L, and Bert-F scores drop more significantly as the target speedup increases. While our approach can achieve substantial speedups, the trade-off in task performance is more pronounced, as the hidden state saturation is reached earlier.

In conclusion, our method consistently demonstrates an ability to decrease computational demands across all datasets and model sizes, effectively determining the hidden state saturation point. The impact on task performance, as measured by Bleu, Rouge-L, and Bert-F scores, varies depending on the specific dataset. However, in all instances, our method shows a favorable balance between speedup and task performance, reaching a $2\times$ speedup with almost no degradation in all cases. This balance can be effectively exploited as *our approach adeptly handles practical challenges like batching and KV caching while maintaining a controlled and predictable computational budget*.

## 3.3 COMPARISON TO OTHER METHODS

In order to benchmark `SkipDecode`, we have adapted two concepts from the CALM framework to function on decoder-only models. The CALM method was introduced for encoder-decoder architectures, and heavily leverages this architecture difference, as we see when we apply the method to decoder-only architectures. In both cases we use OPT 1.3b as the base model. For comparison, we train a multi-layer exit network with two different exit policies. For training, we following the method outlined in Schuster et al. (2022), where a single model head is trained to exit from each layer. The initial checkpoint, data, and hyperparameters (effective batch size, learning rate, warm-up steps) match the `SkipDecode` models, and we noted that when switching from standard to multi-layer loss, the performance of the full network is retained. The first approach for comparison is simply exiting at a fixed layer from this network for every token, which is similar to an early termination method with truncation. Notably, this policy supports batching and KV Caching.

The second method uses the same model, with an additional application of the CALM's hidden state saturation exit policy (we'll refer to this as CALM-DEC). The adaptive hidden state saturation policy on this network has the standard disadvantages of a non-fixed policy for both batching and computation/time estimates. In addition, its performance degrades strongly with increasing speedups especially on larger decoder-only models for the following reason. Note that this policy has no additional learned hyperparameters beyond an exit threshold applied to the difference in state from one layer to the next. We sweep across this threshold to measure the performance-computation tradeoff of this method for decoder-only architectures. While CALM can be adapted for decoder-only architectures, this method imposes a limitation on the batch size to just one, precluding batching and KV Caching. Consequently, the model must 'back-fill' all KV values for previous tokens as required (in this case, by projecting the last known hidden state at that layer), which adds significant systems overhead. The worst case computational cost of this approach is equivalent to the full network cost.

We measure speedup using the analogous measurement, where the CALM-DEC method uses all layers for the prompt (forward pass) and speedup is measured on all tokens generated after the prompt. The results from CALM-DEC show significant degradation for decoder-only models as opposed to the original application of CALM to T5 encoder-decoders. This is in part due to how KV backfill affects the prompt encoding, which is extremely important for these tasks. When applied to decoder-only architectures, the full hidden state of the prompt is no longer accessible to generated tokens that exit early (unlike when the encoder gives the same representation to each token in an encoder-decoder architecture as in Schuster et al. (2022)). One of the motivating reasons for us embarking on this work was noticing this drawback in the CALM method's inability to retain a faithful encoding of the prompt for the generation function in decoder-only architectures, and the resulting loss in performance, especially for highly varying dynamic exit points.

| Speedup | E2E | | | Reddit-TLDR | |
|---|---|---|---|---|---|
| | SkipDecode | Multi-layer | CALM-DEC | SkipDecode | Multi-layer |
| 1 | 67.6 | 68.7 | 68.7 | 27.3 | 26.3 |
| 2 | 67.9 | 65.7 | 35.8 | 27.5 | 17.2 |
| 3 | 68.2 | 61.5 | 32.1 | 25.1 | 12.7 |
| 4 | 66.8 | 50.8 | 27.7 | 21.3 | 7.9 |
| 5 | 66.3 | 46.7 | 22.8 | 19.3 | 6.5 |

Table 4: Performance comparison between `SkipDecode`, Multi-layer, and CALM-DEC.

As can be observed from Table 4, `SkipDecode` exhibits a superior performance over other approaches. This is demonstrated by the notably less degradation in task performance across both datasets as the speedup factor increases. This showcases the robustness of our method against increasing speedup.

## 4 RELATED WORK

**Model compression:** There has been extensive research in model compression to develop techniques to improve the inference efficiency of large language models (LLMs). One of the most prominent lines of work leverage knowledge distillation (KD) Hinton et al. (2015) to train smaller student

models with faster inference using representations from LLMs as teachers like hidden states and attention states Jiao et al. (2019); Sun et al. (2020). Another line of work in model compression use quantization (Gong et al., 2014), low-precision training and network pruning (Han et al., 2016), parameter sharing and factorization to reduce the memory footprint as well as network latency (Gupta & Agrawal, 2022; Treviso et al., 2022). Notably most of the above research in model compression has focused on encoder-only models for natural language understanding tasks.

**Early-exit:** In contrast to the above works that use static computation i.e. the same computation being invoked on every input, we focus on *adaptive compute* with variable computation for different parts of the input. Existing adaptive computation techniques primarily rely on early-exit strategies Zhu (2021); Zhou et al. (2020); Xin et al. (2020); Liu et al. (2020); Li et al. (2021); Hou et al. (2020) where a token in the input learns to exit from different layers of the network. Similar to the works in KD, most of these techniques were developed for encoder-only models like BERT Devlin et al. (2019) for natural language understanding (NLU) tasks. In contrast to NLU tasks that requires processing of the sequences as a whole, generation tasks are more complex given their autoregressive nature for token-by-token generation. A recent work, CALM Schuster et al. (2022) study token-level early exit strategies for generation tasks in terms of what confidence measure to use; how to connect sequence-level constraints to local per-token exit decisions; and how to attend back to missing hidden representations due to early exits in previous tokens. However, similar to all the prior early-exit works, CALM suffers from some major practical blockers related to batching (only supporting a batch size of 1) and KV caching which are widely used to speedup inference in practice. Further, the worst-case scenario for all such exit-strategies (e.g., exit point closer to the top layer for any token) can lead to using the full network resulting in unpredictable system load and inconsistent throughput. To address these challenges, we develop `SkipDecode` that supports non-trivial batching and KV caching, a key component Yan et al. (2021) for efficient inference, as well as guarantees a predictable computational load with no surprises.

## 5 LIMITATIONS AND FUTURE DIRECTIONS

`SkipDecode` addresses pivotal issues like batching and Key-Value (KV) caching, inherently incompatible with existing token-level early exit strategies. However, the introduction of the decaying policy has a limitation. As the generation progresses and samples in the batch finish their computations, new samples can be included in the batch only if their current position matches the remaining elements' positions. Therefore, our method does not naturally support the 'infinite loop' inference mode.

In preliminary experiments, a power law decay function did not yield improvements over the linear decay employed here. Notably, prior research indicate a power law distribution for token exit levels Schuster et al. (2022). Our Oracle exploratory experiments, depicted in Figure 1, corroborate this observation. Investigating alternative decay functions presents an intriguing avenue for future work.

Another promising research direction involves examining the decaying policy's impact on the prompt. In accordance with previous studies, we have utilized the full network for the prompt. Additional speedup gains may be attainable by extending the policy to the prompt and implementing more aggressive decay functions, as mentioned earlier. This could pave the way for more efficient and versatile token-level early exit strategies.

Our study is limited to use cases where the models are finetuned for specific tasks. Whether this type of compression could affect other properties of the model (e.g., emergent capabilities, zero shot) is, however, also an interesting direction of future research.

## 6 CONCLUSIONS

`SkipDecode` bridges the gap between the theoretical benefits of token-level early exits and real-world application requirements. It adeptly addresses practical challenges like batch processing and key-value caching. Moreover, it consistently exhibits the capability to reduce computational requirements by identifying the saturation point of hidden states with a controlled computational budget. This not only enhances efficiency but also fosters a more accessible and sustainable AI ecosystem.

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

## A   APPENDIX

### A.1   DATASET EXAMPLES AND ANECDOTAL RESULTS

| Dataset | Context | Response 2x | Response 5x |
|---------|---------|-------------|-------------|
| E2E | name[Blue Spice], eatType[coffee shop], customer rating[average], near[Burger King] | The Blue Spice coffee shop located near Burger King has been rated average by customers. | Blue Spice is a coffee shop near Burger King. It has an average customer rating and is located near the Burger King. |
| Reddit-TLDR | "SUBREDDIT: r/relationships TITLE: This guy I've been casually dating [18M] doesn't want to have a relationship with me [18F] because he's going to college in the fall POST: Here's a bit of context for y'all: We both met freshmen year in our school's theatre program. At the end of freshman year, I transferred to... | Guy I've been casually dating wants to break up with me because he's going to university in the fall and I have to stay in high school for another year. | Guy I've been dating has been dating for a while, he's going to university in the fall, I'm crushed and don't know how to proceed. |
| CNN-DM | (CNN)The terrorist group Al-Shabaab has claimed an attack on Garissa University College in eastern Kenya, in which many people have been killed and still more taken hostage. The attack is another step in the ongoing escalation of the terrorist group's activities, and a clear indicator that the security situation in East Africa is deteriorating fast. Somalia-based Al-Shabaab has been behind a string of recent attacks in Kenya, the most well-known of them being the massacre at the Westgate Shopping Centre in Nairobi in 2013. Cross-border raids into Kenya by the group, however, date back to 2011. Al-Shabaab incursions triggered a military... | Al-Shabaab claims attack on Garissa University College in Kenya .Attack is another step in the ongoing escalation of terrorist group's activities .Al-Shabaab has been behind a string of recent attacks in Kenya .The group is predominantly driven by the same radical interpretation of the Koran as al-Qaeda . | Al-Shabaab has claimed an attack on Garissa University College in Kenya .Al-Shabaab has been behind a string of recent attacks in Kenya .Al-Shabaab has been behind a string of recent attacks in the region. |

Table 5: Snapshot of dataset and model responses.

## A.2 ROUGE-L VISUAL RESULTS

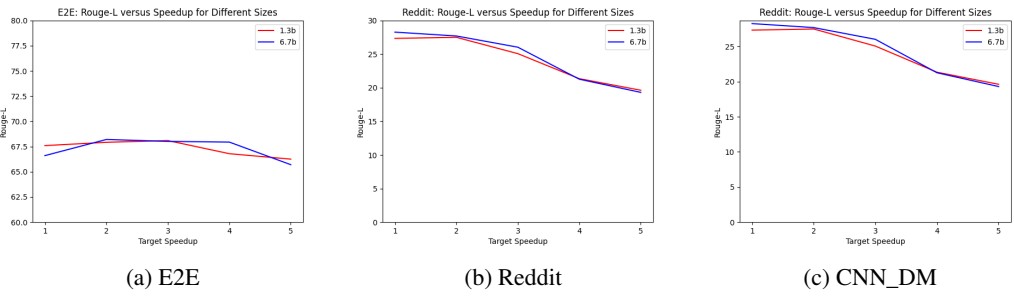

| (a) E2E | (b) Reddit | (c) CNN_DM |

Figure 4: Rouge-L vs inference speedup for 1.3B and 6.7B OPT models. Speedup is computed over base model (1×) that inherently supports batching and KV caching in contrast to prior work considering a weaker base model without batching as reference.

