# OpenReview forum: "SkipDecode: Autoregressive Skip Decoding with Batching and Caching for Efficient LLM Inference"
_ICLR.cc/2024/Conference — Submitted to ICLR 2024_

### Official Review · Reviewer_RFb8 · 2023-10-26

**Soundness:** 2 fair
**Presentation:** 3 good
**Contribution:** 3 good
**Rating:** 6
**Confidence:** 4

**Summary:**

> **TL;DR:** The proposed SkipDecode algorithm achieves significant inference speedups (2x to 5x) across various tasks and LLM model sizes while maintaining negligible performance regression. However, the algorithm has limitations, which are clearly stated and helpful. Addressing my concerns and questions would improve my score.

The paper introduces SkipDecode, a novel token-level early exit strategy designed to enhance the efficiency of autoregressive large language models (LLMs) in natural language generation tasks. The existing token-level early exit methods have limitations when applied to batch inferencing and Key-Value caching, as they require waiting for the last token in a batch to exit, hindering practicality. SkipDecode overcomes these constraints by enabling each token in a batch to exit independently at each sequence position, ensuring a monotonically decreasing exit point. This approach prioritizes computational resources on upper layers, allowing later tokens to benefit from earlier token computations. Experimental results demonstrate that SkipDecode achieves significant inference speedups (2x to 5x) across various tasks and LLM model sizes while maintaining negligible performance regression. This approach not only supports batch processing and KV caching but also identifies the saturation point of hidden states, contributing to a more efficient and sustainable AI ecosystem.

**Strengths:**

* **S.1.** The paper is well written and the illustrations are informative.
* **S.2.** The SkipDecode algorithm is novel, effective and compatible with many existing inference optimization techniques.
* **S.3.** The SkipDecode algorithm outperforms previous algorithms and is evaluated on several datasets with two different LLM sizes.
* **S.4.** The SkipDecode algorithm shows promising speedups and the paper clearly states its limitations.

**Weaknesses:**

* **W.1.** The experiments are conducted solely on a single neural architecture. Providing results on other neural architectures (such as ENcoder-Decoder) would help.
* **W.2.** The SkipDecode algorithm is compared to a single algorithm and the provided results are limited. While the SkipDecode algorithm was evaluated on three different datasets with three different evaluation metrics on each, the comparison evaluation includes only two different datasets with a single metric for each dataset. Adding more comparison results would help.
* **W.3.** The paper was easy to follow, however, I find some information missing. Addressing my questions would improve my score.

Typos:
* saturatedSchuster -->saturated Schuster
* Wile our method --> While our method

**Questions:**

* **Q.1.** How are the min_exit_layer and max_exit_layer chosen?
* **Q.2.** What is the degradation of quality of using batch-wise exit-point function instead of a per example-wise?
* **Q.3.** What is the gain of using Skipping instead of Early-Termination? If the function is monotonic decreasing wouldn't that solve the problem for Early-Termination?
* **Q.4.** How is the speed-up and quality affected by the sequence-length hyperparameter? Why was is chosen to be the median? How would this work in a general chat-bot fashion where the generated sequence length can largely vary?
* **Q.5.** Do the speedups include the prompt computation time?

---

> ### Author Response · Authors · 2023-11-20
>
> **We sincerely appreciate the insightful feedback and thoughtful inquiries provided by the reviewer.** In this response, we aim to comprehensively address the concerns and questions raised, hoping to clarify and enhance the understanding of our research.
>
> **OPT choice and decoder architecture:**
>
> Please refer to the general answer above under "OPT choice and decoder architecture" [here](https://openreview.net/forum?id=bcHty5VvkQ&noteId=rNoNTTEd2m) as this point was raised by another reviewer.
>
> **Compared only with one baseline**
>
> We do not know other work comparable to ours in terms of early exit in the decoding stage. CALM is the closest we are aware of.
>
> Our work is one of the first to focus on early exit strategies in the decoding stage of autoregresive neural networks. There are limited studies that align closely with our approach.
>
> We identified CALM as the closest relevant work to our approach. However, CALM is designed for encoder-decoder architectures. To facilitate a meaningful comparison, we implemented a decoding-only version of CALM, which we believe is the most appropriate baseline given the specific focus of our research. The scarcity of directly comparable works in terms of early exit strategies in decoding stages has limited our ability to conduct a broader comparative analysis with a range of baselines.
>
> The adaptation of CALM to a decoding-only context was necessary to provide at least one viable point of comparison. We acknowledge that this adaptation might not fully capture the nuances of other potential baselines designed explicitly for decoding-only architectures.
>
> While Skipdecode is implemented with the Metaseq framework and therefore we can handle a wide range of large datasets, our implementation of CALM is not as optimized as Metaseq, that's why we limited the comparison to fewer datasets and sizes.
>
> **Q.1. How are the min_exit_layer and max_exit_layer chosen?**
>
> Our approach to selecting min_exit_layer and max_exit_layer is driven by the target speedup, such as 2x. A given target speedup will give us different min_exit_layer and max_exit_layer configurations based on Figure 3 (Section 2.3). We determine these configurations through hyperparameter search on the E2E dataset and then apply the same configurations across other datasets. We touched on this in Section 3.1 (second paragraph) of our manuscript, **we will revise this section to provide a clearer and more explicit explanation of our method for choosing these layers.**
>
> **Q.2. What is the degradation of quality using batch-wise exit-point function instead of a per example-wise?**
>
> In our case there is no degradation per se as it is a position wise exit. In the SkipDecode, we employ a position-wise exit strategy, meaning that tokens at the same position in different elements of the batch exit simultaneously. Our method maintains consistency across batch elements, regardless of the size of the batch.
>
> **Q.3. What is the gain of using Skipping instead of Early-Termination? If the function is monotonic decreasing wouldn't that solve the problem for Early-Termination?**
>
> The key benefit of using Skipping over Early-Termination, even with a monotonic decreasing function, lies in how the transformer architecture processes information layer by layer. In Early-Termination, later tokens might only attend to intermediate hidden states, potentially missing out on crucial information processed in subsequent layers. Skipping ensures that the full computation performed by earlier tokens is utilized. This concept is illustrated in the 'Computational Efficiency' section of Figure 2 in our paper.
>
> **Q.4. How is the speed-up and quality affected by the sequence-length hyperparameter? Why was is chosen to be the median? How would this work in a general chat-bot fashion where the generated sequence length can largely vary?**
>
> The sequence-length hyperparameter, when other variables are held constant, influences the slope of the decaying function (see Figure 3), but it doesn't significantly impact the overall speedup. The choice of sequence length depends on the dataset, as our framework focuses on Fine Tuning and specialization (refer to the [general answer](https://openreview.net/forum?id=bcHty5VvkQ&noteId=PYRP8JGGUt)). Therefore, it is not ideally suited for general chat-bot applications. The sequence length and prompt size will ultimately depend on the dataset training set statistics. We selected the median prompt size during training to allow the model to adapt naturally to the typical prompt sizes found within the dataset's variance (Section 2.3, last paragraph; Section 3.1, 4th paragraph).
>
> **Q.5. Do the speedups include the prompt computation time?**
>
> No, the speedups reported in our study do not include the prompt computation time, only the decoding. **We will clarify it more. We will ensure to make this clearer in the revised version of the manuscript.**

---

### Official Review · Reviewer_rBkT · 2023-10-31

**Soundness:** 3 good
**Presentation:** 3 good
**Contribution:** 3 good
**Rating:** 6
**Confidence:** 4

**Summary:**

The authors claim that there is a problem with early exit methods like CALM as they create K,V cache issues. The authors propose a simple solution - predefine exit location for each token based on the sequence length. The strategy is based on the claim "later tokens are easier to generate than the first tokens".

**Strengths:**

1. The approch is simple. Directly applicable without a requiring a lot of low level implementation.
2. Is the loss per token a valid proxy for "hardness" ?

**Weaknesses:**

1. I am still a bit unconvinced that fixing a schedule for early exit is good approach for reducing the computation. To an extent won't a better approach be do it based on the context rather than forcing it.
2. The second is the validaty of the claim that"KV cache" generation is a massive bottleneck. Yes I agree it might makeup for unpredictable tail latencies, but can authors do an analysis of illustrating the problem.
3. I understand the authors perspective of performing fine-tuning. However, do authors think it is a viable approach in real world, especially as the model sizes keep increasing. How would authors go about creating a dataset which reperesent real world examples.

4. I am having a hard-time understanding Table-4. I have spent close to 15 minutes trying to understand the Table and reading text around. There are no lables, no descriptions what the numbers mean and how do you go about fixing speedup. Please improve the presentation.

All in all, I think I like the idea. However, I am still somewhat on the fence about this paper. I understand the problem about K,V cache with CALM. However, wouldn't just generating K,V cache solve that problem as well. How much proportion of time is regeneration of K,V cache is something I would like to understand.
Looking forward to reading the authors response. Thank you for your great work.

**Questions:**

Please see the weakness section

---

> ### Author Response · Authors · 2023-11-20
> **Response (Part 1)**
>
> **We sincerely appreciate the insightful feedback and thoughtful inquiries provided by the reviewer.** In this response, we aim to comprehensively address the concerns and questions raised, hoping to clarify and enhance the understanding of our research.
>
> **Position-wise schedule:**
>
> You raise a valid point about the potential benefits of a context-based approach over a fixed schedule.
>
> In an ideal scenario, as implemented in methods like CALM, decisions for early exits would indeed be made on a token-by-token basis. However, while theoretically advantageous, this approach faces practical limitations in batched scenarios. The computational theoretical gains presented in methods like CALM are only realized when the batch size is limited to one.
>
> Our method seeks to address this limitation by implementing position-based exits. While a context-driven approach at the batch level is conceptually appealing, it introduces significant complications. Each token in a batch might require a different exit point due to its unique context. For example, a token in the first element of a batch might need an early exit, whereas a token in the last element might benefit from a later exit. This variability can negate computational gains and introduce uncertainty in terms of computational cost.
>
> To ensure efficiency and predictability, our approach focuses on having the entire batch exit at the same position. This exit strategy across the batch ensures that the computational gains are both effective and predictable. In this context, fine-tuning becomes a crucial aspect of our method as it allows us to optimize the exit points for different positions, balancing the trade-off between computational efficiency and the contextual nuances of each token.
>
> **KV cache analysis:**
>
> Generative inference in LLMs often employs a KV Cache mechanism to enhance generation speed. In the causal decoder used by these models, attention for a token is calculated based on preceding tokens, leading to repetitive recalculations for each token generation. KV cache mitigates this by storing and reusing previously computed Key/Value vectors, thereby avoiding redundant computations at the cost of increased memory usage. Without KV caching, for each new token all KV representations for previous tokens, should be recomputed every time.
>
> [1] conducted an experiment with GPT-2 on a Tesla T4 GPU (details available in [1]) to quantify the impact of KV caching. The results were compelling:
>
> With KV Caching: 11.885 ± 0.272 seconds (for the generation of 1000 new tokens)
> Without KV Caching: 56.197 ± 1.855 seconds
>
> This demonstrates the significant efficiency gains offered by KV caching. Our simple method, via skipping, allows the use of KV caching in early exit scenarios and, even more, as it progressively reduces layer usage, it decreases both KV requirements and overall memory footprint.
>
> The memory consumption of KV cache increases rapidly as the model size and generation length increase, drastically increasing the pressure of on-device memory [2], in some scenarios even surpassing the model size [4]. When memory usage exceeds GPU capacity, the generative inference of LLMs typically resort to offloading and the KV cache loading ends up dominating the inference cost. According to [3] inference with multihead attention incurs significant memory capacity and bandwidth costs to store and load the KV cache, and these costs can dominate the rest of the inference at large batches or long context lengths.
>
> Our approach aims to improve both sides of the coin, utilizing KV cache to speed up the process in the context of early exit while minimizing its memory footprint. As [2] and [3] state, dealing with KV caching is critical for advancing efficient generative inference in LLMs.
>
> [1]: "Transformers KV Caching Explained, Lages, 2022" ( https://medium.com/@joaolages/kv-caching-explained-276520203249)
>
> [2]: Model Tells You What to Discard: Adaptive KV Cache Compression for LLMs, Ge et. al., 2023
>
> [3]: Efficiently Scaling Transformer Inference, Pope et. al., 2023.
>
> [4]: Scissorhands: Exploiting the Persistence of Importance Hypothesis for LLM KV Cache Compression at Test Time, Liu et. al, 2023
>
> **Role of Fine-Tuning in SkipDecode:**
>
> Please refer to the general answer above under ["Role of Fine-Tuning in SkipDecode"](https://openreview.net/forum?id=bcHty5VvkQ&noteId=PYRP8JGGUt) title as this point was raised by another reviewer.

---

> ### Author Response · Authors · 2023-11-20
> **Response (Part 2)**
>
> **KV Cache in Early Exit:**
>
> Regarding the specific question about computing the cache for early exit methods. In CALM, and similar early exit methods, generating the K,V cache for tokens exiting at different layers presents some challenges. This is primarily due to the architectural deviations these methods introduce in the standard Transformer model. For example, if a token exits beyond the layer reached by previous tokens, CALM attends to the last computed layer for those tokens. While it is technically feasible to generate the cache in such scenarios, this process necessitates additional logic and careful handling to ensure a closer alignment with the original Transformer architecture.
>
> In contrast, SkipDecode maintains the standard layer-by-layer attention mechanism of the Transformer. This adherence to the conventional architecture means that K,V caching can be utilized straightforwardly, without necessitating low-level implementation adjustments or architectural decisions.
>
> The core advantage here is that SkipDecode retains the efficiency of the Transformer's attention mechanism, avoiding the computational overhead of on-the-fly cache recomputation that early exit methods might incur if they are to adhere to the Transformers original architecture (Figure 2).
>
> **Table 4:**
>
> Table 4 compares SkipDecode with 2 baselines derived from the CALM paper at different levels of Speedup. Speedup reflects the various sizes of the network according to each approach. Multilayer: reflects the number of layers used monolithically, while CALM-DEC reflects the exit points at each token level. The numbers reflect the accuracy in each dataset.
>
> **Sorry for the inconvenience, we will expand the section to include proper descriptions in the revised manuscript.**
>
> Thank you for your feedback regarding Table 4. We apologize for any confusion caused by the presentation of the data.
>
> Table 4 is designed to compare the performance of our proposed system, SkipDecode, against two baseline approaches adapted from the CALM framework. These baselines are a multi-layer exit network and CALM-DEC, a variation tailored for decoder-only models like ours.
>
> The 'Speedup' column in the table represents different levels of computational efficiency achieved by each method. For the Multi-layer approach, 'Speedup' indicates the number of layers used in a monolithic fashion, essentially a measure of how early the model exits. In the case of CALM-DEC, it reflects the dynamic exit points at each token level, determined by the network's hidden state saturation.
>
> The subsequent columns show the performance of each approach on two datasets, E2E and Reddit-TLDR, at various speedup levels. The performance is measured in terms of accuracy, with the numbers provided being percentage scores.
>
> **Upon reviewing your feedback, we realize the need for a more comprehensive explanation of these concepts in the manuscript. **
>
> We hope this clarification addresses your concerns. We will ensure the revised manuscript provides a clearer and more detailed explanation of Table 4.
>
> **Is the loss per token a valid proxy for "hardness"?**
>
> Our intuition is that, the loss per token, which measures the discrepancy between the predicted and actual outputs, can be indicative of the difficulty the model encounters in accurately predicting the correct answer at each token position. This interpretation aligns with the principle of loss functions where higher loss values typically reflect greater challenges in model prediction (and therefore require greater adjustments).
>
> The context within which the model operates plays a significant role. For instance, in sequence-based tasks, predictions made early in a sequence may be more challenging due to limited preceding context, as opposed to later predictions where the model has more contextual information. This observation is consistent with the findings in the CALM paper (specifically Figure 2.a), where the impact of perturbations at different sequence positions was analyzed. The model found it more challenging to recover from perturbations occurring early in the sequence than those towards the end.
>
> In our work, we use loss per token as an indicative measure, but we also acknowledge the complexity of interpreting this metric as a standalone indicator of 'hardness'. Future research could further refine this approach, perhaps by integrating additional metrics or considering the contextual factors that influence model predictions.

---

### Official Review · Reviewer_HbUe · 2023-11-01

**Soundness:** 3 good
**Presentation:** 2 fair
**Contribution:** 2 fair
**Rating:** 5
**Confidence:** 4

**Summary:**

This work investigates token-level early exit for large language models. Existing approaches are an ill fit for batch inference and KV cache. To address these two challenges, the authors propose two designs: a shared exit point for every token in a batch at each sequence position for batching; a monotonic decrease in exit point for KV cache. The authors evaluate their method on three generation datasets using the OPT model.

**Strengths:**

This work investigates an important problem, and provides a practical design for the batching serving setting.

**Weaknesses:**

-> This work only considers the finetuning setting, whereas LLM is particular interesting for its in-context learning ability.

-> The authors only discussed established direction such as distillation and quantization, but no recent works on compressing LLM nor efficient inference of LLM in Section 4.

Lin, Ji, et al. "AWQ: Activation-aware Weight Quantization for LLM Compression and Acceleration." arXiv preprint arXiv:2306.00978 (2023).

Xiao, Guangxuan, et al. "Smoothquant: Accurate and efficient post-training quantization for large language models." International Conference on Machine Learning. PMLR, 2023.

Frantar, Elias, et al. "Gptq: Accurate post-training quantization for generative pre-trained transformers." arXiv preprint arXiv:2210.17323 (2022).

Sheng, Ying, et al. "FlexGen: High-Throughput Generative Inference of Large Language Models with a Single GPU." (2023)

Liu, Zichang, et al. "Deja vu: Contextual sparsity for efficient llms at  inference time." International Conference on Machine Learning. PMLR, 2023.

Zhang, Zhenyu, et al. "H_2 O: Heavy-Hitter Oracle for Efficient Generative Inference of Large Language Models." arXiv preprint arXiv:2306.14048 (2023).

Liu, Zichang, et al. "Scissorhands: Exploiting the Persistence of Importance Hypothesis for LLM KV Cache Compression at Test Time." arXiv preprint arXiv:2305.17118 (2023).


-> Minor: I believe if use \citep can put the citation inside bracket, which will make the pdf much easier to read.

**Questions:**

What is special about E2E?  Why E2E seems to have a significantly better performance? I think this will help us understand when this method works.

---

> ### Author Response · Authors · 2023-11-20
>
> **We sincerely appreciate the insightful feedback and thoughtful inquiries provided by the reviewer.** In this response, we aim to comprehensively address the concerns and questions raised, hoping to clarify and enhance the understanding of our research.
>
> **Role of Fine-Tuning in SkipDecode:**
>
> Please refer to the general answer above under ["Role of Fine-Tuning in SkipDecode"](https://openreview.net/forum?id=bcHty5VvkQ&noteId=PYRP8JGGUt) title as this point was raised by another reviewer.
>
> **Missing angle on efficient LLM inference:**
>
> The reviewer is right and we are thankful for your insightful feedback regarding the coverage of recent works on LLM compression and efficient inference in our manuscript. We acknowledge that incorporating recent advancements in this field would significantly strengthen our discussion and argument, especially concerning KV cache management during LLM inference.
>
> In light of your suggestions, we realize the importance of emphasizing our method's ability to manage KV caches efficiently. Our approach not only facilitates KV caching during early exit scenarios but also effectively reduces the KV cache size, thereby diminishing the memory footprint. This point, although present in our initial submission, was not highlighted sufficiently. We appreciate that your comment has brought this to our attention, and even [1] suggested by the reviewer helped us argue in favor of efficiently dealing with KV caches in other reviewer response.
>
> **We will incorporate a more detailed discussion of these recent works in the revised version of our paper. Specifically, we plan to:**
> 1. Discuss the relevance of these recent methods in the context of our work, particularly focusing on how they complement and contrast with our approach;
> 2. Explicitly highlight how our method addresses some of the challenges posed by LLM compression and efficient inference;
> 3. Strengthen the section on KV cache management, underscoring its significance in efficient LLM inference, as your suggestion and the reference [1] have prompted us to consider.
>
> We believe that these additions will improve the overall quality and comprehensiveness of our manuscript.
>
> Thank you once again for your constructive feedback.
>
> [1] *Specializing Smaller Language Models towards Multi-Step Reasoning, Fu et. al., 2023*
>
> **E2E Dataset:**
>
> Our approach is particularly effective in scenarios where generation tasks are extensive and the scope is well-defined. This makes it well-suited for applications such as summarization and translation.
>
> The E2E dataset is also an ideal fit. It focuses on transforming structured data into coherent verbalizations, usually involving concise prompts. For instance, a typical prompt in the E2E dataset might be: {"prompt": "name[The Dumpling Tree], eatType[restaurant], food[Italian], priceRange[high]"}, with the corresponding response: "The Dumpling Tree is an Italian restaurant with high prices." This dataset requires the model to convert structured, attribute-based information into a fluid narrative form – a task that aligns seamlessly with the strengths of our method.
>
> The reason our method shows significant performance improvements with the E2E dataset can be attributed to several factors:
> 1. The E2E dataset's well-defined nature and focused scope enable our model to specialize effectively via fine-tuning.
> 2. Our method demonstrates its strength in maintaining quality up to the point of hidden state saturation. In the case of the E2E dataset, this threshold lies within the manageable size of the model. This indicates that our method can effectively handle the dataset's tasks without requiring a very large network. The good numbers in all speedups indicate that the method is able to avoid degradation up to the hidden state saturation point.
>
> In summary, the E2E nature of its tasks play into the strengths of our method, allowing for more efficient language generation. This is reflected in the performance metrics we observed.

---

### Official Review · Reviewer_zrR5 · 2023-11-09

**Soundness:** 2 fair
**Presentation:** 3 good
**Contribution:** 3 good
**Rating:** 5
**Confidence:** 4

**Summary:**

This work proposes SkipDecode, an early-exit method for speeding up the inference of autoregressive models. This method works by setting a fixed schedule for skipping earlier layers depending on the number of generated tokens. Authors validate the performance of SkipDecode by testing it on OPT models for several generative tasks, showing speedups over prior methods such as CALM.

---

Post-rebuttal update: thanks to the authors for their response! Since they have not followed up with the experiments they proposed to conduct, I am keeping my current score.

**Strengths:**

* The work proposes an original way to simplify early-exit adaptive generation techniques that addresses their shortcomings. The method is conceptually simple yet efficient in practice.
* Overall, the paper is well-written and the key contributions are clear.
* The empirical results for studied models and datasets appear promising: in some cases, there are negligible accuracy degradations even with a 5x speedup.

**Weaknesses:**

* For a work that claims practical performance speedups of deep learning inference, it should be important to comprehensively evaluate the real-world increase in speed compared to the baselines. However, I have found that part of the experiments to be missing multiple crucial details, for example, the type of hardware used to run the experiments, the batch size during generation, and the metric that was used to obtain the numbers for true speedup (was it latency, throughput, or something else?). Also, apparently there is no real-world speedup comparison between SkipDecode and CALM-DEC.
* The choice of datasets could also be more comprehensive: currently, 2/3 problems are related to summarization with quite long prompts, and another is structure-to-text conversion. To give a broader view of whether SkipDecode performs reliably well across different problems, it would be useful to include tasks with different input-output relations and sequence length (for example, machine translation experiments from the CALM paper)
* There are quite a few typos in the submission: for example, "figure 2" -> "Figure 2" at the bottom of page 3, "e2e dataset" -> "E2E dataset" and "figure 3" -> "Figure 3" on page 6, "SkipDecodemodels" -> "SkipDecode models" on page 8.
* Lastly, I think that the OPT family of models is not fully representative of the architecture variations used today (for example, LLaMA models with multi-query attention), and the current findings about embedding saturation might not transfer to larger or more recent models.

**Questions:**

* If I understood correctly, each of the experiments you ran involved finetuning the model on a target dataset. However, in practice, model providers might often serve a single model for many different applications: for example, this means that the average prompt/response lengths can vary dramatically. Is it possible to extend SkipDecode to such a scenario?
* Which hardware did you use to run your experiments and how did you measure the true speedup?

---

> ### Author Response · Authors · 2023-11-20
>
> **We sincerely appreciate the insightful feedback and thoughtful inquiries provided by the reviewer.** In this response, we aim to comprehensively address the concerns and questions raised, hoping to clarify and enhance the understanding of our research.
>
> **Speedup measurement and Hardware details:**
>
> **We will add hardware information and real time for the SkipDecode experiments.**
>
> As hardware we use a single node with 16xV100 GPUs.
>
> With respect to the baseline, real time might not be the best metric as our implementation is by no means optimized as it is the OPT Metaseq implementation.
>
> However, our aim here is to compare different configurations (2x, 3x, 4x, 5x) against the full network cost (1x) both in performance and speedup.
>
> **We will include detailed information about the hardware used for running our experiments. Additionally, we will provide real-time measurements for the SkipDecode experiments to offer a more tangible understanding of the speedups achieved.**
>
> While we understand the importance of real-time metrics, we would like to clarify that our baseline CALM-DEC implementation is not fully optimized. SkipDecode and CALM-DEC are implemented on different frameworks. Therefore, comparing real-time metrics directly might not accurately reflect the efficiency of the underlying method.
>
> We followed the metric originally proposed by CALM which is the number of layers used, which is a good proxy for time and resources needed. Same as CALM, the True Speedup in table 3 is measured as the Total Number of Layers in the network by the Average Number of Layers used per token. Difference with CALM is that we can do batching so this measure includes a batched setting (i.e., batches of arbitrary size), while CALM restricts the analysis to a scenario with a batch size equal to 1.
>
> However, as our aim here is to compare different configurations (2x, 3x, 4x, 5x) against the full network cost (1x) both in performance and speedup. We can provide real-time metrics for different configurations of SkipDecode.
>
> We believe these additions and clarifications will comprehensively address the concerns raised and provide a clearer picture of the practical performance speedups achievable by SkipDecode. We appreciate the opportunity to enhance our work based on your feedback.
>
> **More Diverse Datasets:**
>
> Please refer to the general answer above under ["More Diverse Datasets"](https://openreview.net/forum?id=bcHty5VvkQ&noteId=ok0kucxlVL) title as this point was raised by another reviewer.
>
> **OPT choice and decoder architecture:**
>
> Please refer to the general answer above under ["OPT choice and decoder architecture"](https://openreview.net/forum?id=bcHty5VvkQ&noteId=rNoNTTEd2m) title as this point was raised by another reviewer.
>
> **Role of Fine-Tuning in SkipDecode:**
>
> Please refer to the general answer above under ["Role of Fine-Tuning in SkipDecode"](https://openreview.net/forum?id=bcHty5VvkQ&noteId=PYRP8JGGUt) title as this point was raised by another reviewer.
>
> **Adapting parameters to different datasets:**
>
> We indeed fine-tune for each dataset (see ["Role of Fine-Tuning in SkipDecode"](https://openreview.net/forum?id=bcHty5VvkQ&noteId=PYRP8JGGUt)). To adapt for each dataset, we use the statistics of the training set. To handle varying prompt lengths we train the network using the median prompt size (Section 3.1) which has worked well in our experiments.
>
> Regarding the prompt length again we set the max_len based on our training set, and if a datapoint in test is longer than that it will be assigned the min_layer for each token beyond the max_len parameter (Section 3.1).
>
> * Thank you very much for the typos we will fix it in the reviewed version

---

### Author Response · Authors · 2023-11-20
**Role of Fine-Tuning in SkipDecode [Reviewers 1, 2, 3]**

Our approach is by design tailored to tackle model specialization scenarios and thus fine-tuning plays a pivotal role. By paying the price of decreased generic ability, it is possible to generate smaller specialized models at a fraction of the size (and cost) of larger counterparts with comparable performance; this has recently shown very effective in Instruction Tuning settings for specific tasks [1, 2]. Our method enables further gains into specialized models by addressing practical considerations in early exit strategies.

Fine Tuning is indeed deemed as an effective way to specialize LLMs[3,4], and even providers such as OpenAI or Azure now give the possibility to fine tune LLMs of various sizes for specific data.

Applicability to Varied Applications: Model specialization is a promising direction especially in relatively simpler tasks as summarization or translation, that required long generation steps. While it's true that LLM providers often serve a single model for multiple applications, the specialization strategy focuses on efficiency and targeted performance. By fine-tuning for specific tasks like summarization, translation, or coding, we aim to reduce the computational load and costs associated with generating large numbers of tokens. This doesn't preclude the use of a more generalized model for other applications; rather, it offers a complementary approach where specialized models handle easier tasks that are token-intensive and hence more costly in a general-purpose setting.

In-Context Learning vs. Fine-tuning: While in-context learning is a hallmark of LLMs, fine-tuning offers distinct advantages, particularly in terms of performance optimization for specific tasks. Our method leverages the strengths of fine-tuning to achieve higher efficiency and effectiveness in these specialized tasks. This is not to diminish the value of in-context learning but to highlight the role of fine-tuning in enhancing model performance for certain applications.

Scalability and Real-World Viability: Addressing the concern about the increasing size of models, our approach advocates for a more sustainable and cost-effective strategy. By fine-tuning smaller models for specific tasks, we can achieve comparable performance with significantly reduced resource requirements. This is particularly relevant as the field moves towards larger, more complex models.

In summary, our fine-tuning approach towards model specialization is not only a viable but a necessary strategy in the evolving landscape of LLMs. It balances performance with efficiency, particularly in tasks that are resource-intensive when handled by general-purpose models.

[1] Instruction Tuning with GPT-4, Peng et. al., 2023

[2] Orca: Progressive Learning from Complex Explanation Traces of GPT-4, Mukherjee et. al., 2023

[3] Challenges and Applications of Large Language Models, Kaddour et. al., 2023

[4] Specializing Smaller Language Models towards Multi-Step Reasoning, Fu et. al., 2023

---

### Author Response · Authors · 2023-11-20
**OPT choice and decoder architecture [Reviewers 1, 4]**

We chose a decoder-only architecture primarily due to its prevalence in recent research and applications. This choice was not driven by a belief that our method is specifically tailored to the OPT model family. On the contrary, our intention was to design a method easily applicable to decoder transformer architectures, benefiting from their inherent characteristics.

Another significant factor in our decision was the efficiency offered by the Metaseq codebase. This framework enabled us to conduct our experiments more effectively, especially given our limited computational resources. At the time of starting our research, Metaseq was a promising tool, though it unfortunately did not gain as much traction as anticipated.

We are indeed planning to expand our research to encompass a wider range of models. This expansion requires migrating to a different infrastructure, which is a substantial effort in terms of time and resources. We are committed to this and believe it will greatly enrich our research's applicability and relevance.

We understand the concern that conducting experiments solely on a single model might limit the generalizability of our findings. Our initial focus was to establish a robust foundation within the constraints of our resources. Moving forward, we aim to provide results on models as suggested. This will not only enhance the robustness of our research but also broaden its applicability across various model families.

---

### Author Response · Authors · 2023-11-20
**More Diverse Datasets [Reviewers 1, 4]:**

Our method is specifically tailored for specialization scenarios and long text generation. It is aimed to work well in contexts where the generation phase is extended, and the task has a well-defined scope. This aligns perfectly with tasks like summarization and translation. The E2E dataset, a well-defined task, which focuses on verbalizing structured data with relatively short prompts, was also identified as an ideal fit for our method.

We acknowledge the zrR5 reviewer's  point regarding the absence of a translation dataset. Our decision was influenced by the fact that OPT, the model we employed, was not explicitly trained for multilingual capabilities. However, acknowledging OPT's limited yet potential utility in multilingual scenarios, as highlighted by its authors, we agree that incorporating a translation dataset could be beneficial. **In response to your suggestion, we will conduct an experiment using the WMT dataset, and report the results here.**

For summarization, we chose the CNN_DM dataset for its focus on professionally written articles and the Reddit dataset to address the challenges of summarizing more colloquial and noisy text prevalent in social media.

We are also open to integrating additional datasets like XSum, which offers a different flavor of summarization. **If you have any specific datasets in mind that would align well with our research scope, we are more than willing to consider them** and include our findings in our revised submission. In the interim, **we will proceed with experiments on both WMT and XSum datasets and report them here.**

We hope this clarifies our dataset selection rationale and demonstrates our commitment to comprehensively evaluating our method's efficacy across varied scenarios.

---

### Meta-Review · Area_Chair_foTq · 2023-12-20

**Metareview:**

All reviewers found this paper interesting but have several concerns about the experiment setting and the choice of baselines.

**Justification For Why Not Higher Score:**

It would be great if the papers can provide more end to end experiments and more comprehensive comparisons with baselines.

**Justification For Why Not Lower Score:**

N/A

---

### Decision · Program_Chairs · 2024-01-16

Reject